# Food insecurity and psychological stress among migrants and refugees in high-income countries: Protocol for a systematic review and meta-analysis

Resom Berhe[1,2]*, Amit Arora[1,3,4,5,6], Kanchana Ekanayake[7], Kingsley E. Agho[1,3,8]

**1** School of Health Sciences, Western Sydney University, Penrith, NSW, Australia, **2** Department of Health Education and Behavioural Sciences, Institute of Public Health, College of Medicine and Health Science, University of Gondar, Gondar, Ethiopia, **3** Translational Health Research Institute, Western Sydney University, Penrith, NSW, Australia, **4** Discipline of Child and Adolescent Health, Sydney Medical School, Faculty of Medicine and Health, The University of Sydney, Westmead, NSW, Australia, **5** Oral Health Services, Sydney Local Health District and Sydney Dental Hospital, NSW Health, Surry Hills, NSW, Australia, **6** Health Equity Laboratory, Campbelltown, NSW, Australia, **7** University of Sydney Library, Camperdown, NSW, Australia, **8** Faculty of Health Sciences, University of Johannesburg, Johannesburg, South Africa

* resom.berhe86@gmail.com

**Data Availability Statement:** Deidentified research data will be made publicly available when the study is completed and published.

## Abstract

### Background

Numerous studies have established a correlation between Food Insecurity (FI) and diminished physical and psychological well-being. However, there is a lack of knowledge regarding this association among migrant and refugee populations. Migrants and refugees face difficulties, such as a lack of legal clarity and employment opportunities, which can exacerbate FI and psychological well-being issues. Therefore, the main goal of this study is to conduct a systematic review of the existing body of literature, followed by a meta-analysis of the results, where appropriate, to examine how common food insecurity is among migrants and refugees living in high-income countries and whether it might be linked to mental distress.

### Methods

The inclusion of studies will be contingent upon providing quantitative data on migrants and refugees in developed countries. This criterion encompasses all pertinent observational study designs and has been published in English. The review will specifically include cross-sectional, case-control, and cohort studies that utilize standardized and validated measurement tools for food insecurity (e.g., Food Insecurity Experience Scale) and psychological stress (e.g., 21-item Depression Anxiety and Stress Scale (DASS-21)), excluding non-standard or non-validated measures. A systematic search will be conducted across electronic databases such as Ovid Medline, Embase, SCOPUS, and Web of Science, containing peer-reviewed journal papers in health, psychology, and social sciences from January 1, 2008, to the present for relevant studies. Prevalence estimates will be generated using meta-analysis with a 95% CI, 95% prediction intervals, and $I^2$ statistics for heterogeneity.

**Funding:** The author(s) received no specific funding for this work.

**Competing interests:** The authors have declared that no competing interests exist.

The studies incorporated in the review will be analysed using meta-analysis, as deemed suitable for the characteristics of the data obtained.

## Discussion

This study has practical implications for policies and interventions, offering insights for evidence-based initiatives targeting food security and mental health among migrants and refugees, thus improving resource allocation.

**PROSPERO registration number:** CRD42024525690.

## Introduction

### Description of the condition

The issue of Food Insecurity (FI), which refers to the inability to access sufficient quantities of safe and nutritious food due to physical, social, or financial constraints, is a pressing global concern acknowledged by the Food and Agriculture Organization (FAO) [1]. While traditionally associated with developing countries, FI has emerged as a significant challenge in affluent nations such as Australia, Europe, the United Kingdom, Canada, and the United States [2]. Its impacts extend beyond marginalised groups and affect broader populations, resulting in significant economic and societal consequences.

The global economic and financial crisis of 2008 and subsequent events, such as the ongoing COVID-19 pandemic, conflicts or war leading to fuel and food price shocks, and climate-related disasters, have exacerbated the challenges faced by FI. These events have increased living costs, increased food insecurity, and heightened humanitarian needs [3, 4]. Migrants and refugees have been disproportionately affected by these developments [3]. While transitioning to new regions, humanitarian migrants encounter two primary obstacles: food shortages and psychological stress [5–7]. These challenges not only impact individuals but also have a broader impact on host societies, emphasising the importance of prioritising the health and well-being of migrant populations for sustainable development [8].

Recent data from FAO (2020) [9] indicate that approximately 30% of the world's population faces FI, with this percentage on an upward trajectory. Notably, refugees and migrants are disproportionately vulnerable to FI due to multiple factors that span economic, cultural, and social domains. Limited economic opportunities often lead to financial insecurity, restricting their ability to buy adequate food. Additionally, lacking social support networks reduces their access to community assistance programs. The challenge of navigating a new cultural environment—such as finding familiar or culturally acceptable foods and adapting to new dietary practices—further exacerbates their food insecurity [5, 10, 11]. Studies reveal that asylum seekers and recently resettled refugees in affluent nations experience FI at rates ranging from 8 to 20 percent [2]. For instance, a study in Brisbane, Australia, found that nearly a quarter of the population grapples with FI, with socioeconomically disadvantaged areas exhibiting a higher prevalence due to lower household incomes and a greater burden of unfavourable health conditions [12]. Similarly, research conducted in Western Australia illuminates the link between food insecurity and sociodemographic factors among newly arrived immigrants, with the majority reporting inadequate access to food [13, 14]. This is not unique to Australia.

FI poses a substantial challenge for migrants in Norway, where a comprehensive analysis indicates that 93% of refugees experience FI, compared to only 3% of Norwegians [15]. When

we look at the situation in the United States of America, mainly through a survey conducted in Massachusetts in July 2020 that reached out to immigrant communities in 16 different languages, it becomes evident that food insecurity is a significant challenge. The findings revealed that 59% of the surveyed families reported insufficient food. This issue was even more acute among families with at least one undocumented member, where the figure rose to 78%, starkly highlighting the depth of food insecurity challenges faced by immigrant populations [16]. Those experiencing FI are often characterised by lower incomes, reliance on food aid, difficulties associated with grocery shopping, and language barriers. These hardships result in adverse consequences such as restricted food choices, micronutrient deficiencies, small portions, and skipped meals [17]. However, it is important to note that despite extensive research on host populations, there is a significant gap in understanding FI among migrants not residing in reception facilities, particularly young males travelling alone as refugees and asylum seekers–a population that has received insufficient scholarly attention to date [18].

Simultaneously, humanitarian migrants face a pressing mental health crisis, necessitating focused efforts to address and reduce mental health inequalities within this vulnerable group [19]. Experiences of migration and refugeehood increase the risk of physical and mental health issues during the challenging process of adapting to new countries and cultures. Empirical evidence has consistently shown that migrants are more susceptible to mental health problems than native-born individuals [20–23]. For instance, a preliminary review of Australian data indicated that 16 percent of migrants were diagnosed with severe mental illness, while 31 percent exhibited symptoms of post-traumatic stress disorder (PTSD) [24]. Additionally, a meta-analysis of 35 studies revealed that depression and anxiety rates are approximately 20% for migrants and 40% for refugees. Of significant concern is the persistence of these symptoms over many years after resettlement, raising intergenerational health concerns [25, 26]. Moreover, an extensive review of data across 13 nations spanning North America, the Pacific, Asia, and Europe underscores the higher prevalence of depression, anxiety, and somatic disorders among migrants compared with the native population, with rates two to five times greater [27] than native residents.

## Why the review is important

Existing research underscores the heightened vulnerability of migrants and refugees to both food insecurity and psychological stress [28–31]. Nevertheless, while studies exploring the relationship between migration and food insecurity have predominantly reviewed qualitative research, the association between food insecurity and psychological stress remains inadequately elucidated. Moreover, existing studies are often limited to specific subsets of migrants or focus on immigrants from particular countries, leading to inconclusive and conflicting findings owing to methodological disparities [32–34]. The focus on high-income countries is strategically chosen to highlight the unique barriers that migrants and refugees face in these settings, where, paradoxically, higher economic stability does not always translate to food security for vulnerable populations [6]. Unlike low- and middle-income countries, where food insecurity might stem more directly from supply-side issues, in HICs, food insecurity often results from economic and social barriers, such as lack of access to social services, discriminatory policies, and high living costs [35]. This review aims to provide evidence that can guide policymakers in HICs to develop tailored interventions that address these specific barriers and enhance food security and mental well-being among migrant and refugee populations [36].

The study period cutoff of 2008 aligns with the onset of the global financial crisis, which profoundly impacted global migration patterns and economic conditions [37]. Post-2008, there was a marked increase in food insecurity due to economic recessions, austerity measures,

and rising unemployment rates in many high-income countries [38]. Furthermore, migration policies in response to economic crises often became more restrictive, exacerbating vulnerabilities for migrants and refugees [39]. By focusing on studies conducted from 2008 onwards, we ensure the relevance and comparability of the data in light of these significant global socio-economic shifts [40]. This approach also allows the review to capture the compounded effects of subsequent global events, such as the COVID-19 pandemic, which further intensified food insecurity and psychological stress among migrant populations [41].

Furthermore, there is an urgent need to thoroughly investigate how food insecurity affects various health outcomes for migrants in wealthy nations. Most public health research focuses on the nutritional components of food insecurity. This nutritional bias results from a combination of factors, including the clear biological relationship between food and nutrition and the anthropometric nature of many national and regional measures of food insecurity. This paradigm, which emphasises physical and nutritional health, may mask the effects of food insecurity on other health and well-being indices that anthropologists such as Howard and Millard (1997) [42] and Scheper-Hughes (1992) [43] have so eloquently detailed. This highlights the pressing need for additional research to resolve these disparities and produce a more thorough understanding of the association between FI and psychological stress, particularly for immigrants residing in HIC.

**Objectives.** This systematic review and meta-analysis aim to estimate the pooled prevalence of food insecurity and psychological stress and their association among migrants and refugees residing in high-income countries based on World Bank-classified regions.

## Proposed methodology

### Protocol and registration procedures

This systematic review and meta-analysis protocol was officially registered with the PROSPERO network, with the registration number CRD42024525690. This study will follow a meticulously structured protocol encompassing the design, search methodology, analysis, and dissemination of findings, as outlined in the systematic review procedures recommended by the PRISMA (Preferred Reporting Items for Systematic Reviews and Meta-Analyses) guidelines [44]. The adherence to the PRISMA Protocols (PRISMA-P) checklist [45], which details the essential elements for reporting in systematic reviews and meta-analyses, is demonstrated in the supplementary material provided (S1 File).

### Eligibility criteria

Studies will be selected based on the criteria in Table 1. The PECOS elements form the basis of the eligibility requirements.

### Exposure

Various factors, including determinants such as food insecurity, have a significant impact on the occurrence of psychological stress. FI is a prominent worldwide issue that is distinguished by the inability to secure a sufficient and safe supply of nutritious food, which results from physical, social, and economic constraints. Additional terms that characterise food insecurity include poverty, deprivation, insufficiency, and hunger. Food insecurity is assessed by employing the Food Insecurity Experience Scale (FIES) survey, following the guidelines established by the Food and Agriculture Organization (FAO) [46].

**Table 1. Study selection eligibility criteria.**

| PICOS | Inclusion | Exclusion |
|---|---|---|
| **P (Population)** | Migrant and refugee adults aged > 18, including those with various migration statuses, such as recently resettled refugees, undocumented individuals, and permanent residents. | Studies focusing exclusively on a specific subgroup unrelated to immigrants or the broader population |
| **E (Exposure)** | Food insecurity. Other terms for food insecurity include Food hunger, deprivation, insufficiency, and poverty. | Food-secured population group |
| **Comparison (C)** | Not applicable | |
| **Outcome (O)** | Psychological distress (Depression, Anxiety, and Stress) | Accounts and Experiences not Directly Concerning Food |
| **S (Study Types)** | studies using quantitative and mixed methods, such as all observational research, including cross-sectional, case-control, and cohort studies. | Policy Documents Discourse analysis Editorials Conference abstracts |
| **Study Period** | In the past fifteen years, published (from January 2008 till present) The year 2008 was chosen as the start date since it was the year that the global financial crisis started, and since then, there has been an increase in the number of people who are food insecure [4]. | Articles published prior to January 2008. Studies only collected before 2008 |
| **Setting** | High/very high human development index nations (HIC) | Non-HIC |
| **Language** | Studies written in English | language other than English |

## Outcome

Studies describing experiences and accounts of how FI affects psychological stress will be included. Studies that measured psychological stress using the well-established and validated 21-item Depression, Anxiety, and Stress Scale (DASS 21) (Ali et al., 2021) [47] will be included. We will also consider studies that used the Kessler Psychological Distress Scale (K10) where appropriate, as it is another commonly used measure of psychological stress in similar populations. Note that this scale is not intended for use as a diagnostic tool for psychological distress; its primary purpose is to assess the degree of significant characteristics associated with depression, anxiety, and stress.

## Search strategy and information sources

The present systematic review will encompass all scholarly articles published. The systematic review will follow the PRISMA 2020 methodology Page et al. (2021) [44] outlined to ensure methodological rigour. SCOPUS, Embase, Medline (OVID), and Web of Science will be the electronic databases used for the review, spanning the years 2008–2024. These databases were selected because of their relevance to food insecurity and mental distress and their extensive coverage of published research articles and journals. The search strategy will be developed in collaboration with an experienced librarian from The University of Sydney. An example of a search strategy used in Ovid MEDLINE is provided as a S2 File. The search terms included a combination of "migrant," mental stress, "food insecurity," and "food security" and their synonyms. Researchers will initiate communication with the authors of research articles with incomplete data. This will consist of instances when a pertinent conference abstract has been identified, where the authors will be approached to request full-text articles. Search efforts will also include the grey literature database for both reports and unpublished studies. The search

criteria will be broadly and universally applied across all disciplines to reduce the risk of rejecting pertinent publications during the initial assessment of the titles and abstracts.

## Study selection

The selected research papers will be imported into the citation management tool Covidence (accessible at www.covidence.org), where duplicate entries will be systematically eliminated. The selected articles will be screened, and relevant studies will be selected for further consideration by two independent reviewers. If there is a disagreement, the decision to include an article will be determined after a discussion and sharing of perspectives among the research team members. The search results from various databases will be consolidated using EndNote version 20.1. Subsequently, redundant entries will be eliminated. The remaining studies will undergo a comprehensive screening process based on their titles and abstracts to exclude irrelevant findings that fail to match the predetermined inclusion criteria. Comprehensive reports will then be evaluated for subsequent compliance investigations, adhering to predetermined standards for records that have passed the initial screening process.

## Data abstraction

Upon identifying the relevant literature to be included in the review, we will employ a data abstraction form to extract information concerning various aspects of the studies, such as the study's time frame, design, sampling method, data collection method, and participant demographics. The reviewer will autonomously complete the data abstraction spreadsheets, which will then be cross verified by other reviewers. The data abstraction form will be utilised to extract information from publications, including details on study outcomes such as exposure and outcome description, the methods employed for measurement, and the resulting findings. If available, preliminary data on the number of participants in both the exposure and result groups will also be extracted from the articles. The descriptions of exposure and outcome will be condensed and streamlined to create a table of study results accessible and conducive to comparison. A pilot study will be conducted using a limited literature sample carefully chosen for inclusion in the review to guarantee a thorough compilation of pertinent and accurate data. The abstraction tool will be tested and adjusted as needed during this process. The summary will group studies with similar populations and recorded outcome measures for quality assessment and suitable data synthesis.

## Quality assessment

We will employ the Joanna Briggs Institute (JBI) Prevalence Critical Appraisal tool to evaluate the calibre of the publications included in the study [48]. These tools adhere to the established standards for evaluating the quality of systematic reviews and meta-analyses. In all instances, two reviewers (RB and KE) will conduct the assessment autonomously and assign a final evaluation to determine the quality of the article. Discrepancies will be effectively addressed through the process of deliberation and consensus, and other reviewers (KA and AA) will be consulted if necessary. Quality evaluation data will be a descriptive tool in systematic food insecurity reviews.

## Data synthesis

This study will use quantitative data synthesis methods to understand research findings comprehensively. Results will be classified by setting, considering unique participant and environmental features. The total pooled prevalence of the association between food insecurity and

psychological stress will be estimated using a weighted inverse variance random-effects model [49]. To account for potential discrepancies in the overall prevalence estimate, subgroup analyses will be conducted based on high-risk groups, the geographical area of the included studies, and the study design. Subgroup analyses will be conducted in specific high-risk subgroups, such as unaccompanied minors, recently resettled refugees ($<$ 5 years versus longer term), and undocumented migrants. These subgroups are more vulnerable to food insecurity and psychological stress due to specific socio-economic and legal challenges. By identifying and analysing these subgroups separately, the review aims to provide more nuanced insights into the varying levels of vulnerability and the specific needs of these populations. Subgroup analyses will also be conducted based on regional differences, such as those in North America, Europe, and Australia/New Zealand, to address contextual variability across high-income countries. Contextual variables, including immigration policies, social safety nets, and economic conditions, will be considered in the analysis. For instance, differences in welfare policies between Nordic countries and the United States could result in varying levels of food insecurity among migrants and refugees. By conducting these regional subgroup analyses, the review will provide a more nuanced understanding of how these contextual differences influence the relationship between food insecurity and psychological stress. Additionally, to assess the robustness of the findings, sensitivity analyses will be performed by excluding studies with a high risk of bias or those that use different definitions or measurements of food insecurity and psychological stress. These statistical methods will ensure a rigorous data evaluation, leading to more reliable and valid conclusions. The assessment of publication bias among studies will involve using a funnel plot and Egger's regression test [50]. Statistical tests will be performed using the STATA software.

## The status and timeline of the study

The review is ongoing, and we expect it to be completed. The results will be reported within 12 months.

## Ethics and dissemination

The review is using secondary data, so ethical approval is not required. Review findings will be disseminated by publication in theses, peer-reviewed academic journal articles, conferences, and policy and practice workshops organised by the Western Sydney University Centre for Translational Research in Public Health and circulated to the general public and stakeholder groups using social media.

## Discussion

This study detailed protocol for systematic and meta-analysis of food insecurity and psychological stress and their association among migrants and refugees residing in high-income countries based on World Bank-classified regions. This review will be the first systematic review of research from quantitative studies on the association between FI and mental stress. The COVID-19 pandemic, conflicts, and climate-related calamities have had a substantial influence on food insecurity (FI) globally, as has the global economic slump of 2008. Not only have these events increased economic volatility, but they have also made particular groups more vulnerable to food shortages and psychological stress—most notably, migrants and refugees in high-income nations. The growing number of people with refugee backgrounds residing in these nations has directed attention toward the particular factors impacting their food security, recognizing the presence of particular obstacles such as language, access to work, and social support [51, 52].

The 2008 financial crisis had a complex effect on food insecurity among disadvantaged populations in high-income nations, according to a preliminary investigation [51]. The global economic crisis of this century had a profound impact on people's quality of life and mental health, especially for vulnerable populations such as refugees and migrants. The relationship between financial strains like unemployment and lower income and psychological stress started to show during this time of unstable finances. Although governments were able to keep food supply sufficient, there was considerable variation in their ability to guarantee these vulnerable populations access to food and stabilize prices [52]. Sen (1981) [53] argued that a person's access to food, which is essential for security, depends on a variety of circumstances, including income. Béné (2020) [54] suggests that the crisis's diverse effects on food insecurity are likely impacted by the resilience of national food systems. This demonstrates the significance of demand-side factors, such as food price and accessibility, even in situations where the food supply is adequate, as the Global Food Security Report (2009) [55] emphasizes.

The results validate the predictions made by certain organizations that macroeconomic downturns worsen food insecurity and put nutritional well-being at risk [56]. They may also lead to the long-term deterioration of nutritional strategies that are important for guaranteeing food access and other essential items associated with food security [57]. To properly comprehend these complicated dynamics, a more comprehensive assessment across high-income countries is recommended, taking into account their varying degrees of development and the particular difficulties experienced by migrants and refugees [58]. This is based on the preliminary analysis conducted. To adequately address these particular issues and guarantee food security for disadvantaged groups in the face of continuous global crises, focused actions and policies supported by systematic research—such as quantitative systematic reviews and meta-analyses—are required.

The principal challenge that we will encounter in this review will stem from the considerable variability in the methodologies and data quality across different studies. Such discrepancies introduce potential biases that could skew the outcomes of our analysis. To mitigate these issues and enhance the reliability of our findings, it is imperative to establish and adhere to a rigorous protocol. This protocol was a critical safeguard against arbitrary decision-making and methodological inconsistencies, ensuring a standardized data collection, analysis, and interpretation approach. Moreover, while the DASS-21 is a widely used and validated tool for measuring psychological stress, we recognize the limitations of relying solely on self-reported data. To address this, we will prefer studies that use additional clinical assessments or validated tools, such as the General Health Questionnaire (GHQ) or clinician-administered PTSD Scale (CAPS), to cross-verify the findings. We will compare the consistency of findings from studies using self-reported measures and those incorporating clinical assessments, discussing any discrepancies arising from methodological differences."

Academically, this research is significant as it focuses on an underrepresented group of migrants and refugees in affluent nations. It sheds light on individuals' specific difficulties with food insecurity and psychological stress by employing rigorous methodologies such as meta-analysis for empirical precision. Furthermore, this study has practical implications for policies and interventions, offering insights for evidence-based initiatives targeting food security and mental health among migrants and refugees, thus improving resource allocation. It is a valuable resource for scholars, educators, and students interested in migration, food security, and mental health. By conducting this review and reporting the meta-analysis, this study aims to fill a substantial knowledge gap regarding the impact of food insecurity on psychological stress among migrants and refugees in high-income countries, with far-reaching consequences for policy, practices, and migrants' overall well-being.

## Conclusion

This systematic review and meta-analysis aims to compile available data on the association between food insecurity and psychological stress among migrants and refugees residing in high-income countries. Our findings will provide valuable insights into the prevailing trends and significant gaps in current research, which will be used to guide specific interventions and policy improvements. The main focus of this project is to improve the support systems and resources provided to disadvantaged populations, with the aim of promoting better mental health outcomes and enhancing their overall quality of life.

## Supporting information

**S1 File. Checklist of the Preferred Reporting Items for Systematic Reviews and Meta-Analyses Protocols (PRISMA-P) statement adapted for a systematic review protocol.**
(DOCX)

**S2 File. Search strategy for MEDLINE.**
(DOCX)

## Acknowledgments

This study is part of the first author's thesis for a doctoral dissertation with the School of Health Sciences at Western Sydney University, Australia.

## Author Contributions

**Conceptualization:** Resom Berhe, Amit Arora, Kingsley E. Agho.

**Data curation:** Kanchana Ekanayake.

**Methodology:** Resom Berhe, Amit Arora, Kingsley E. Agho.

**Writing – original draft:** Resom Berhe, Amit Arora, Kingsley E. Agho.

**Writing – review & editing:** Resom Berhe, Amit Arora, Kingsley E. Agho.

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
