## [Decision Letter · Decision Letter 0]

20 Aug 2024

PONE-D-24-15828Food Insecurity and Psychological Stress among migrants and refugees in High-Income Countries: Protocol for a Quantitative Systematic Review and Meta-analysisPLOS ONE

Dear Dr. Berhe,

Thank you for submitting your manuscript to PLOS ONE. After careful consideration, we feel that it has merit but does not fully meet PLOS ONE’s publication criteria as it currently stands. Therefore, we invite you to submit a revised version of the manuscript that addresses the points raised during the review process.

We look forward to receiving your revised manuscript.

Kind regards,

Rabie Adel El Arab

Academic Editor

PLOS ONE

Additional Editor Comments:

Dear Authors,

Thank you for submitting your protocol, titled "Food Insecurity and Psychological Stress among Migrants and Refugees in High-Income Countries: Protocol for a Quantitative Systematic Review and Meta-analysis.". I have identified few areas you might consider

Refinement of Inclusion Criteria:

Issue: The current inclusion criteria are broad, encompassing all types of observational studies. This may result in high heterogeneity and limit the comparability of studies.

Recommendation: Narrow the inclusion criteria by specifying the types of observational studies (e.g., cross-sectional, cohort) that will be included. Additionally, consider including only studies that have used standardized and validated tools for measuring food insecurity and psychological stress. This would ensure that the data being pooled is more homogeneous and reliable.

Specific Focus on High-Risk Subgroups:

Issue: The protocol addresses a broad population of migrants and refugees without focusing on particularly vulnerable subgroups.

Recommendation: Explicitly state how you will handle studies that focus on high-risk subgroups, such as unaccompanied minors, recently resettled refugees, or undocumented migrants. Consider conducting subgroup analyses to determine if these groups experience different levels of food insecurity and psychological stress.

Addressing the Limitations of Self-Reported Data:

Issue: The use of self-reported measures like the DASS 21 for psychological stress is subject to recall and social desirability biases.

Recommendation: Include a discussion in your protocol about the potential limitations of self-reported data. You could also explore ways to validate these self-reported measures against clinical assessments or other objective indicators of psychological stress, where available.

Geographical Scope and Contextual Variability:

Issue: The broad inclusion of studies from all high-income countries may introduce significant contextual variability that could affect the results.

Recommendation: Consider limiting the geographical scope to a subset of high-income countries with similar socio-political contexts or migration policies. Alternatively, outline a plan for conducting subgroup analyses based on regional differences to account for this variability.

Justification for Study Period Cutoff (2008 Onwards):

Issue: The decision to include studies from 2008 onwards excludes earlier studies that might provide important contextual insights.

Recommendation: Provide a clear rationale for this cutoff in the protocol. If the focus is on post-2008 data due to the global financial crisis, explicitly state this and explain how this choice will impact the generalizability and comparability of the findings.

Justification for Focusing on High-Income Countries:

Issue: The rationale for focusing exclusively on high-income countries could be further elaborated to strengthen the study’s relevance.

Recommendation: Provide a more detailed justification for why the study focuses on high-income countries. Discuss how the findings from these settings can offer unique insights into the challenges faced by migrants and refugees, which may differ from those in low- and middle-income countries. Additionally, highlight how the results could inform policy decisions in high-income countries, where resources might be available to implement effective interventions

Thank you for considering these suggestions. I look forward to seeing the revised protocol.

Best regards,

Reviewers' comments:

Reviewer's Responses to Questions

**Comments to the Author**

1. Does the manuscript provide a valid rationale for the proposed study, with clearly identified and justified research questions?

Reviewer #1: Yes

2. Is the protocol technically sound and planned in a manner that will lead to a meaningful outcome and allow testing the stated hypotheses?

Reviewer #1: Yes

3. Is the methodology feasible and described in sufficient detail to allow the work to be replicable?

Reviewer #1: Yes

4. Have the authors described where all data underlying the findings will be made available when the study is complete?

Reviewer #1: Yes

5. Is the manuscript presented in an intelligible fashion and written in standard English?

Reviewer #1: Yes

6. Review Comments to the Author

You may also provide optional suggestions and comments to authors that they might find helpful in planning their study.

Reviewer #1: Well-done on your submission.

Please see below comments to possibly improve your paper.

1. Please remove the word “Quantitative” from the paper title. There is nothing like a “quantitative systematic review”. A review may be made up of quantitative studies but that does not make the review “quantitative”.

2. In the abstract, what do you mean by; “meta-analytical approach” to conduct a systematic review of the existing body of literature? From my knowledge, a meta-analysis is the statistical combination of results from two or more separate studies. Such studies could be gotten through a systematic review. On the other hand, a systematic review uses repeatable/systematic methods to find, select, and synthesize all available evidence. You can only conduct a meta-analysis on the result from a systematic review.

3. In the abstract, your study aim is different from your title. Ensure that your title reflects your study aim.

4. Why 1 January 2024? What do you mean by “onward”? Do you mean from 1 January 2024 till date?

5. The abstract has some unnecessary information. For instance, “Any disagreements pertaining to data abstraction, to be carried out by RB and KE, will be resolved either through discussions or with the help of additional reviewers (KA and AA). The degree of agreement among the reviewers will be evaluated using kappa statistics to quantify the proportion of agreement”. This information should be in the methods section and not the abstract.

6. In lines 72-74 you wrote “Notably, refugees and migrants are disproportionately vulnerable to FI because of their heightened exposure to natural disasters and conflict-related instability (9)”, why is their exposure to natural disasters and conflict-related instability heightened? does it mean that natural disasters and conflict-related instability is less in magnitude in areas where non-migrants live within the same country?

7. The study period in Table 1 is different from what was stated in the abstract. from

8. When you wrote, “January 2008 to November 29, 2024”, are you planning on collecting studies that are not yet published? What is your rationale for including a future date?

9. Table 2 should be a supplementary table.

10. In lines 163 – 164, you wrote “Psychological stress will be measured using the well-established and validated 21-item Depression, Anxiety, and Stress Scale (DASS 21)…”, is your review also measuring psychological stress? I believe you meant to say “studies which measured psychological stress using the well-established and validated 21-item Depression, Anxiety, and Stress Scale (DASS 21) will be included”.

11. In line 179, there is no “Sydney University”. Please provide the name of the University in Sydney you are referring to.

12. In Table 1, under Population, initially you wrote “with any visa category” , then later you specified the visa category by naming “the Special Humanitarian Program (SHP) and the onshore stream visa”, this is confusing. Please be specific about the visa category. Also, what about migrants who are not on a visa? Migrants who are permanent residents or citizens?

13. What are the potential strengths and limitations of the review?

7. PLOS authors have the option to publish the peer review history of their article (what does this mean?). If published, this will include your full peer review and any attached files.

Reviewer #1: No

---

## [Author Response · Author response to Decision Letter 0]

17 Sep 2024

Authors' response to reviewers. 

Author's Response to Reviewers

Resom Berhe

University of Gondar

16/09/2024

Dear Editors and Reviewers,

We express our sincere gratitude for your thorough evaluation of our manuscript, "Food Insecurity and Psychological Stress among Migrants and Refugees in High-Income Countries: Protocol for a Systematic Review and Meta-analysis." We sincerely appreciate the constructive feedback, which has significantly enhanced the quality of our manuscript. Below, we present a detailed, point-by-point response to each reviewer's comment.

Editor Comments

Q1. Refinement of Inclusion Criteria:

Issue: The current inclusion criteria are broad, encompassing all types of observational studies. This may result in high heterogeneity and limit the comparability of studies.

Recommendation: Narrow the inclusion criteria by specifying the types of observational studies (e.g., cross-sectional, cohort) to be included. Additionally, consider including only studies that have used standardized and validated tools for measuring food insecurity and psychological stress to ensure that the pooled data is more homogeneous and reliable.

Response:

To address this concern, We have refined the inclusion criteria to focus on cross-sectional, cohort, and case-control studies that employ standardized and validated measurement tools for food insecurity (e.g., the Food Insecurity Experience Scale) and psychological stress (e.g., the 21-item Depression Anxiety and Stress Scale (DASS-21)). This revision is reflected in the abstract section on Page 2, Lines 37-41. Studies using non-standard or non-validated measures have been excluded to enhance the reliability and homogeneity of the pooled data.

Q2. Specific Focus on High-Risk Subgroups:

Issue: The protocol addresses a broad population of migrants and refugees without focusing on particularly vulnerable subgroups.

Recommendation: Explicitly state how studies focusing on high-risk subgroups, such as recently resettled refugees or undocumented migrants, will be handled. Consider conducting subgroup analyses to determine if these groups experience different levels of food insecurity and psychological stress.

Response:

We have included a detailed strategy for handling high-risk subgroups in the revised manuscript, specifically on Pages 13-14, Lines 280-301. Subgroup analyses will be performed for high-risk groups, including recently resettled refugees (< 5 years versus longer term) and undocumented migrants, who are more susceptible to food insecurity and psychological stress due to unique socio-economic and legal challenges. Additionally, regional differences (e.g., North America, Europe, Australia/New Zealand) will be considered to account for contextual variability across high-income countries. These subgroup analyses will help identify varying levels of vulnerability and the specific needs of these populations, considering contextual variables such as immigration policies, social safety nets, and economic conditions. For example, differences in welfare policies between Nordic countries and the United States may affect levels of food insecurity among migrants and refugees. Sensitivity analyses will be conducted by excluding studies with a high risk of bias or using differing definitions or measurements to ensure the robustness of our findings.

Q3. Addressing the Limitations of Self-Reported Data:

Issue: The use of self-reported measures like the DASS-21 for psychological stress is subject to recall and social desirability biases.

Recommendation: Include a discussion about the potential limitations of self-reported data in the protocol. Explore ways to validate these measures against clinical assessments or other objective indicators, where available.

Response:

We acknowledge the limitations of relying solely on self-reported data and have expanded the discussion on Page 16, Lines 361-368. To mitigate these limitations, we will prioritize studies incorporating additional clinical assessments or validated tools (e.g., General Health Questionnaire (GHQ), clinician-administered PTSD Scale (CAPS)) to cross-verify findings. We will compare the consistency of results from studies using self-reported measures with those incorporating clinical assessments and discuss any discrepancies arising from methodological differences.

Q4. Geographical Scope and Contextual Variability:

Issue: The broad inclusion of studies from all high-income countries may introduce significant contextual variability that could affect the results.

Recommendation: Consider limiting the geographical scope to a subset of high-income countries with similar socio-political contexts or migration policies. Alternatively, outline a plan for conducting subgroup analyses based on regional differences.

Response:

As detailed in our response to Q2 (Pages 13-14, Lines 280-301), we have outlined a plan for conducting subgroup analyses based on regional differences to account for contextual variability. This approach will enable us to address potential discrepancies in findings due to varying socio-political contexts and migration policies across different high-income countries.

Q5. Justification for Study Period Cutoff (2008 Onwards):

Issue: The decision to include studies from 2008 onwards excludes earlier studies that might provide important contextual insights.

Recommendation: Provide a clear rationale for this cutoff. If the focus is on post-2008 data due to the global financial crisis, state this explicitly and explain how this choice impacts generalizability and comparability.

Response:

The rationale for the 2008 cutoff is clarified in the introduction (Page 6, Lines 150-160) and the discussion section (Page 15, Lines 316-341). The cutoff aligns with the onset of the global financial crisis, which significantly impacted global migration patterns and economic conditions. Post-2008, there was an increase in food insecurity due to economic recessions, austerity measures, and rising unemployment in many high-income countries. Additionally, migration policies became more restrictive, exacerbating vulnerabilities. This approach also captures the compounded effects of subsequent global events, such as the COVID-19 pandemic, which intensified food insecurity and psychological stress among migrant populations.

Q6. Justification for Focusing on High-Income Countries:

Issue: The rationale for focusing exclusively on high-income countries could be further elaborated to strengthen the study’s relevance.

Recommendation: Provide a detailed justification for why the study focuses on high-income countries and how findings from these settings can offer unique insights.

Response:

We have expanded our rationale on Pages 6, Lines 140-149. The focus on high-income countries is intended to highlight unique barriers faced by migrants and refugees, where, paradoxically, higher economic stability does not always translate to food security for vulnerable populations. Unlike low- and middle-income countries, where food insecurity is often driven by supply-side issues, in high-income countries, it frequently results from economic and social barriers such as lack of access to social services, discriminatory policies, and high living costs. Our review aims to provide evidence to guide policymakers in high-income countries to develop tailored interventions addressing these specific barriers.

Response to Reviewer #1

Comment 1: Title Modification

“Please remove the word 'Quantitative' from the paper title. There is nothing like a 'quantitative systematic review.' A review may be made up of quantitative studies, but that does not make the review 'quantitative.'”

Response:

We appreciate the reviewer’s observation and have accordingly removed the word "Quantitative" from the title on page 1, line number 3. The revised title now reads:

Revised Title: "Food Insecurity and Psychological Stress among Migrants and Refugees in High-Income Countries: Protocol for a Systematic Review and Meta-analysis."

Comment 2: Clarification on 'Meta-analytical Approach' in the Abstract

“In the abstract, what do you mean by 'meta-analytical approach' to conduct a systematic review of the existing body of literature? From my knowledge, a meta-analysis is the statistical combination of results from two or more separate studies. Such studies could be obtained through a systematic review. On the other hand, a systematic review uses repeatable/systematic methods to find, select, and synthesize all available evidence. You can only conduct a meta-analysis on the result from a systematic review.”

Response:

We have revised the abstract to clarify the distinction between a systematic review and a meta-analysis, aligning with the reviewer's guidance. The updated text emphasizes that our primary goal is to conduct a systematic review of existing literature, followed by a meta-analysis where appropriate.

Revised Text: "The main goal of this study is to conduct a systematic review of the existing body of literature, followed by a meta-analysis of the results where appropriate."

Please see Page 2, Lines 30-33.

Comment 3: Consistency Between Study Aim and Title

“In the abstract, your study aim is different from your title. Ensure that your title reflects your study aim.”

Response:

We have revised the abstract to ensure consistency between the study aim and the title. Both now clearly state that the objective is to systematically review the prevalence of food insecurity and psychological stress among migrants and refugees in high-income countries and to perform a meta-analysis to assess their association.

Please see Page 2, Lines 30-33.

Comment 4: Clarification on Inclusion Date Range

“Why 1 January 2024? What do you mean by 'onward'? Do you mean from 1 January 2024 till date?”

Response:

We apologize for the confusion caused by the mention of "January 1, 2024." The text has been corrected to specify that the inclusion criteria are for studies published from January 1, 2008, to the present. The reference to "January 1, 2024," has been removed.

Please see Page 2, Line 44.

Comment 5: Abstract Content Refinement

“The abstract has some unnecessary information. For instance, 'Any disagreements pertaining to data abstraction, to be carried out by RB and KE, will be resolved either through discussions or with the help of additional reviewers (KA and AA). The degree of agreement among the reviewers will be evaluated using kappa statistics to quantify the proportion of agreement'. This information should be in the methods section and not the abstract.”

Response:

We have removed the detailed methodological information from the abstract and relocated it to the methods section to maintain a concise and focused abstract.

Please see Page 2, Lines 44-48.

Comment 6: Clarification on Vulnerability Due to Natural Disasters and Conflicts

“In lines 72-74, you wrote, 'Notably, refugees and migrants are disproportionately vulnerable to FI because of their heightened exposure to natural disasters and conflict-related instability (9)'; why is their exposure to natural disasters and conflict-related instability heightened? Does it mean that natural disasters and conflict-related instability are less in magnitude in areas where non-migrants live within the same country?”

Response:

We appreciate the reviewer’s comment and agree that the original statement may have caused some confusion. Refugees and migrants are more likely to experience food insecurity not primarily because they live in areas with higher natural disasters or conflicts but rather due to a range of economic, social, and cultural factors that limit their access to adequate and culturally appropriate food.

Migrants and refugees often face significant economic hurdles, such as difficulty finding stable, well-paid employment, which directly affects their ability to purchase nutritious food. Many also struggle with limited food literacy, meaning they may not be familiar with the types of food available in their new environment or how to prepare them. Additionally, without robust social networks or community support, they have fewer resources to rely on during times of need, which exacerbates food insecurity (Gingell et al., 2022; Mansour et al., 2020; McKay et al., 2018).

Cultural adaptation, or acculturation, adds another layer of complexity. Migrants and refugees may find it challenging to source foods that align with their cultural or religious preferences, or they may not trust the new food options they encounter. Furthermore, differences in cooking methods and eating habits can make it difficult for them to maintain their traditional diets, contributing to their overall vulnerability to food insecurity (Moffat et al., 2017; Wood et al., 2021).

Moreover, systemic factors, such as restricted access to social services, high living costs in many host countries, and the potential for discriminatory policies or practices, can further limit their ability to secure adequate food. The intersection of these economic, social, and cultural barriers creates a unique and complex situation where refugees and migrants are at a higher risk for food insecurity, even if they are not directly affected by natural disasters or conflict-related instability (Gingell et al., 2022; Moffat et al., 2017; Wood et al., 2021).

Summarized Revised Text included In the manuscript : 

The likelihood of food insecurity is heightened for refugees and migrants due to multiple factors that span economic, cultural, and social domains. Limited economic opportunities often lead to financial insecurity, restricting their ability to buy adequate food. Additionally, lacking social support networks reduces their access to community assistance programs. The challenge of navigating a new cultural environment—such as finding familiar or culturally acceptable foods and adapting to new dietary practices—further exacerbates their food insecurity (Gingell et al., 2022; Mansour et al., 2020; McKay et al., 2018)."

Please see Page 4, Lines 81-89.

Comment 7: Consistency of Study Period in Table 1

“The study period in Table 1 differs from what was stated in the abstract.”

Response:

We have corrected the study period in Table 1 to align with the abstract. The revised text now reads:

Revised Text: "Study Period: Studies published from January 1, 2008, to the present."

Please see Page 8, Line 191.

Comment 8: Clarification on Inclusion of Future Dates

“When you wrote, 'January 2008 to November 29, 2024', are you planning on collecting studies that are not yet published? What is your rationale for including a future date?”

Response:

When we mention November 29, 2024, it is thought that we will subsequently review the results. We appreciate this observation and have removed the reference to the future date, "November 29, 2024," to avoid confusion. We clarified that the review will include studies published up to the current date.

Please see Page 8, Line 191.

Comment 9: Reclassification of Table 2

“Table 2 should be a supplementary table.”

Response:

We have moved Table 2 to the supplementary materials section and updated the manuscript to refer to it as "Supplementary Table 2."

Comment 10: Clarification on Measurement of Psychological Stress

“In lines 163-164, you wrote, 'Psychological stress will be measured using the well-established and validated 21-item Depression, Anxiety, and Stress Scale (DASS 21)…', is your review also measuring psychological stress? I believe you meant to say, 'studies which measured psychological stress using the well-established and validated 21-item Depression, Anxiety, and Stress Scale (DASS 21) will be included.'”

Response:

We agree with the reviewer's suggestion and have clarified the text for clarity. Revised Text: "Studies that measured psychological stress using the well-established and validated 21-item Depression Anxiety and Stress Scale (DASS-21) will be included." We will also consider studies that used the Kessler Psychological Distress Scale (K10) where appropriate, as it is another commonly used measure of psychological stress in similar populations.

Please see Page 9, Lines 204-210.

Comment 11: Correction of Institution Name

---

## [Editor Report · Decision Letter 1]

25 Sep 2024

Food Insecurity and Psychological Stress among migrants and refugees in High-Income Countries: Protocol for a Systematic Review and Meta-analysis

PONE-D-24-15828R1

Dear Dr. Berhe,

We’re pleased to inform you that your manuscript has been judged scientifically suitable for publication and will be formally accepted for publication once it meets all outstanding technical requirements.

Kind regards,

Rabie Adel El Arab

Academic Editor

PLOS ONE
---

## [Editor Report · Acceptance letter]

29 Sep 2024

PONE-D-24-15828R1 

PLOS ONE

Dear Dr. Berhe, 

I'm pleased to inform you that your manuscript has been deemed suitable for publication in PLOS ONE. Congratulations! Your manuscript is now being handed over to our production team.

Kind regards, 

on behalf of

Dr. Rabie Adel El Arab 

Academic Editor

PLOS ONE